# CURRICULUM REINFORCEMENT LEARNING VIA MORPHOLOGY-ENVIRONMENT CO-EVOLUTION

## ABSTRACT

Throughout long history, natural species have learned to survive by evolving their physical structures adaptive to the environment changes. In contrast, current reinforcement learning (RL) studies mainly focus on training an agent with a fixed morphology (e.g., skeletal structure and joint attributes) in a fixed environment, which can hardly generalize to changing environments or new tasks. In this paper, we optimize an RL agent and its morphology through "morphology-environment co-evolution (MECE)", in which the morphology keeps being updated to adapt to the changing environment, while the environment is modified progressively to bring new challenges and stimulate the improvement of the morphology. This leads to a curriculum to train generalizable RL, whose morphology and policy are optimized for different environments. Instead of hand-crafting the curriculum, we train two policies to automatically change the morphology and the environment. To this end, (1) we develop two novel and effective rewards for the two policies, which are solely based on the learning dynamics of the RL agent; (2) we design a scheduler to automatically determine when to change the environment and the morphology. We find these two designs are critical to the success of MECE, as verified by extensive ablation studies. In experiments on two classes of tasks, the morphology and RL policies trained via MECE exhibit significantly better generalization performance in unseen test environments than SOTA morphology optimization methods. Our ablation studies on the two MECE policies further show that the co-evolution between the morphology and environment is the key to the success.

## 1 INTRODUCTION

Deep Reinforcement learning (RL) has achieved unprecedented success in some challenging tasks (Lillicrap et al., 2016; Mnih et al., 2015). Although current RL can excel on a specified task in a fixed environment through massing training, it usually struggles to generalize to unseen tasks and/or adapt to new environments. A promising strategy to overcome this problem is to train the agent on multiple tasks in different environments (Wang et al., 2019a; Portelas et al., 2019; Gur et al., 2021; Jaderberg et al., 2017) via multi-task learning or meta-learning (Salimans et al., 2017; Finn et al., 2017). However, it increases the training cost and the space for possible environments/tasks can be too large to be fully explored by RL. So how to select the most informative and representative environments/tasks to train an RL agent to evolve generalizable skills becomes an critical open challenge. Curriculum learning (Narvekar et al., 2020) for RL aims at developing a sequence of tasks for RL to progressively improve its generalization performance through multiple training stages. However, the curriculum usually relies more on human heuristics, e.g., moving from easy environments to hard ones, chasing the ones with the greatest progress, while lacking sufficient innate incentives from the agent itself to drive the changes towards improving the learning process.

Unlike RL agents that do not actively seek new environments to improve their learning capability, natural species have full motivations to do so to survive in the competitive world, and one underlying mechanism to drive them is evolution. Evolution is a race with the changing environment for every species, and a primary goal is to accelerate its adaptation to new environments. Besides merely optimizing its control policy, evolution more notably changes the morphology of species, i.e., the skeletal structure and the attributes for each part, in order to make them adapt to the environment. In fact, the improvement on morphology can be more critical because there could exist a variety of actions or skills an agent cannot do (no matter how the control policy is optimized) without certain structures, e.g., more than one leg, a long-enough limb, or a 360-degree rotation joint. For RL, we claim that **A good morphology should improve the agent's adaptiveness and versatility, i.e., learning faster and making more progress in different environments.** From prehistoric person to modern Homo sapiens, there is a definite association between the rise of civilization and the Homo sapiens'

optimization of their physical form for improved tool use. Unfortunately, the morphology in many RL researches are pre-defined and fixed so it could be sub-optimal for the targeted environments/tasks and may restrain the potential of RL. Although morphology can be optimized using RL (Sims, 1994; Wang et al., 2019b; Kurin et al., 2021; Yuan et al., 2022) to improve the final performance for a specific task or environment, it was not optimized for the adaptiveness to varying environments. Moreover, instead of re-initializing the control/RL policy after every morphology modification or environment change, the millions of years of evolution demonstrate the effectiveness of continual and progressive learning of the control policy over generations of morphology evolution along with a sequence of varying environments. How to optimize the morphology and control policy of an RL agent for the above goal is still an underexplored problem.

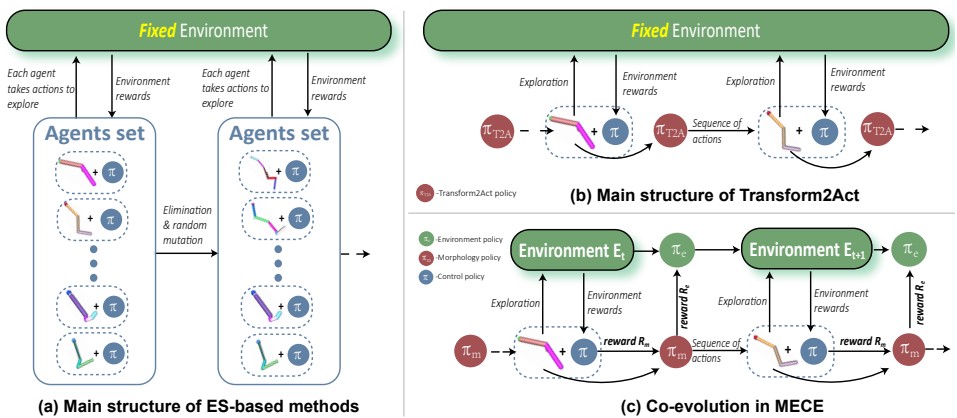

Figure 1: **MECE vs. previous methods.** (a) Evolution strategy (ES)-based method for morphology optimization. It starts from agents with different morphology, eliminates those with poor performance, and then applies random mutation to the survivors. (b) Transform2Act, which trains an RL policy to modify the morphology of an agent in a fixed environment. (c) MECE (ours). We train two policies to optimize the morphology and change the environment to form a curriculum over the course of training a control policy.

Given that the agent has the freedom to improve its morphology and control policy for faster adaptation to different environments/tasks, the remaining question is: what environment can improve this learning process and incentivize the agent to keep finding better morphology (rather than staying with the same one)? The learning environment plays an essential role in RL since it determines the data and feedback the agent can collect for its training. In this paper, we adopt a simple criterion: **A good environment should accelerate the evolution of the agent and help it find better morphology sooner,** which will enhance the agent's adaptiveness and versatility over new environments/tasks.

The interplay between morphology and environment has lasted for billions of years. In the endless game, species which find the appropriate environments to evolve its morphology and policy may survive and keep being improved via adaptation to new environments/tasks. Inspired by this, we propose an *morphology-environment co-evolution (MECE)* scheme that automatically generates a curriculum of varying environments to optimize the agent's morphology and its RL policy so they can be generalized to unseen environments and adapted to new morphology, respectively. In MECE, we train two RL policies to modify the morphology and change the environment, which create a sequence of learning scenarios to train the control policy continuously. This differs from previous work that select the morphology or the environment from a pre-defined set of candidates. Specifically, unlike zeroth-order optimization approaches (Wang et al., 2019b; Hejna et al., 2021; Wang et al., 2018), a morphology policy network can generate modifications based on all the morphology evaluated in diverse environments from history. On the other hand, an environment policy can produce a sequence of progressively modified environments instead of randomly selected or sampled ones (Portelas et al., 2019; Wang et al., 2019a). In Fig. 1, we illustrate the co-evolution process of MECE and compare it with evolution-strategy (ES) based method (Wang et al., 2019b) and Transform2Act (Yuan et al., 2022), which uses RL to optimize the morphology in a fixed environment.

Inspired by the former discussion of "good morphology" and "good environment", we optimize the morphology policy to improve the agent's adaptiveness+versatility and optimize the environment policy to improve the morphology's evolution. To this end, we design two rewards for the two policies based on the learning progress of the control policy, which is estimated by comparing its concurrent

performance on a validation set and its historical trajectories. Therefore, the three policies are complementary and mutually benefit each other: (1) the morphology and environment policies create a curriculum to improve generalizability of the control policy; (2) the environment policy creates a curriculum to optimize the morphology policy; (3) the control policy provides rewards to train the other two policies. In MECE, we train the control policy and automatically determine when to apply the other two policies to modify the morphology or change the environment: they are triggered by the slow progress on the current morphology and environment, respectively. In experiments over different tasks, the morphology and control policy learned through MECE significantly outperforms the existing method in new environments. Moreover, MECE exhibits much faster-learning speed than baselines. We further provide an extensive ablation study to isolate each main component of MECE's contribution and compare it with other options.

## 2 Related Work

**Continuous Design Optimization.** A line of research in the community has studied optimizing an agent's continuous design parameters without modifying its skeletal structure, and they commonly optimize on a specific kind of robots. Baykal & Alterovitz (2017) studies an annealing-based framework for cylindrical robots optimization. Ha et al. (2017); Desai et al. (2017) optimize the design of legged robots by trajectory optimization and implicit function. Applying deep RL into design optimization becomes more popular recently. Chen et al. (2020) uses computational graphs to model robot hardware as part of the policy. CMA-ES (Luck et al., 2019) optimizes robot design via a learned value function. Ha et al. (2017) proposes a population-based policy gradient method. Another line of work (Yu et al., 2019; Exarchos et al., 2021; Jiang et al., 2021) search for the optimal robot parameters to fit an incoming domain by RL. Different from the prior works, our approach learns to find the most general skeleton of an agent while maintaining its continuous design parameters optimization to diverse environments or tasks.

**Combinatorial morphology Optimization.** One closely related line of work is the design of modular robot morphology spaces and developing algorithms for co-optimizing morphology and control (Sims, 1994) within a design space to find task-optimized combinations of controller and robot morphology. When the control complexity is low, evolutionary strategies have been successfully applied to find diverse morphologies in expressive soft robot design space (Cheney et al., 2013; 2018). For more expressive design spaces, GNNs have been leveraged to share controller parameters (Wang et al., 2019b) across generations or develop novel heuristic search methods for efficient exploration of the design space (Zhao et al., 2020). In contrast to task specific morphology optimization, Hejna et al. (2021) propose evolving morphologies without any task or reward specification. Hejna et al. (2021) employ an information-theoretic objective to evolve task-agnostic agent designs.

**Generalization to environments.** Several recent works (Wang et al., 2019a; Portelas et al., 2019; Gur et al., 2021) show that the efficiency and generalization of RL can be improved by training the policy in different environments. By sampling the parameters of environmental features, they Portelas et al. (2019); Wang et al. (2019a) teach the RL agent using a curriculum of various environments. To estimate the distribution of environments, however, requires evaluating the RL policy on numerous sampled environments, which might be expensive and inaccurate for intricate ones. The modified environments may also be unfeasible or too difficult for RL exploration. In MECE, we train an environment policy to change the environments adaptive to the evolved agent's learning progress, and it is possible to regulate the level of difficulty of the modified environments.

## 3 Formulations of the Three Policies in MECE

In MECE, we have three RL policies, a control policy $\pi$ that learns to control the agents of evolved morphology, a morphology policy $\pi_m$ that learns to modify the agent's morphology for better robustness in diverse environments, and an environment policy $\pi_e$ that learns to change the environment to boost the morphology evolution. During the training phase, $\pi_m$ and $\pi_e$ are alternately applied to evolve the agent's morphology and the training environment, respectively. Taking into account that $\pi$ might require various environment steps for training in distinct morphologies or environments, we propose a dynamic time-scaling for $\pi_m$ and $\pi_e$ that is adaptive to $\pi$'s learning process.

**Agent control.** The problem of controlling an agent can be modeled as a Markov decision process (MDP), denoted by the tuple $\{\mathcal{S}, \mathcal{A}, p, r, p\}$, where $\mathcal{S}$ and $\mathcal{A}$ represent the set of state space and action space respectively, $p$ means the transition model between states, and $r$ is the reward function. An agent of morphology $m_i \in M$ in state $s_t \in \mathcal{S}$ at time $t$ takes action $a_t \in \mathcal{A}$ and the environment returns the agent's new state $s_{t+1}$ according to the unknown transition function $p(s_{t+1}|s_t, a_t, m_i)$, along with associated reward $r_t = r(s_t, a_t)$. The goal is to learn the control policy $\pi^* : \mathcal{S} \to \mathcal{A}$ mapping states to actions that maximizes the expected return $\mathbb{E}[\mathcal{R}_t]$, which takes into account reward at future time-steps $J(\pi) = \mathbb{E}[\mathcal{R}_t] = \mathbb{E}\left[\sum_{t=0}^{H} \gamma^t r_t\right]$ with a discount factor $\gamma \in [0, 1]$, where $H$ is the variable time horizon.

**Morphology evolution.** We model the morphology evolution as an MDP problem, denoted by the tuple $\{\mathcal{S}_m, \mathcal{B}, p_m, r_m, \rho\}$. $\mathcal{S}_m$ represents the set of state space, and a state $s_{\alpha t}^m \in \mathcal{S}_m$ at time-step $\alpha t$ is defined as $s_{\alpha t}^m = G_{\alpha t}$, where $G_{\alpha t}$ is the skeleton structure of agent. The action $b_{\alpha t}$ sampled from the set of action space $\mathcal{B}$ can modify the topology of the agent, that has three choices in $\{AddJoint, DelJoint, NoChange\}$. The transition dynamics $p_m(s_{\alpha t+1}^m|s_{\alpha t}^m, b_{\alpha t})$ reflects the changes in state and action. We define the reward function for $\pi_m$ as the average improvement of training $\pi$ with the current evolved morphology in different environments. The reward $r_m$ at time-step $\alpha t$ is denoted as

$$r_{\alpha t}^m = \frac{1}{|E|} \sum_{\theta^E \in E} \left( \mathcal{R}(H, \theta^E, G_{\alpha t}) - \mathcal{R}(H, \theta^E, G_{\alpha t-1}) \right) - \lambda \|a_{\alpha t}^m\|_2^2, \tag{1}$$

note that the first term in Eq. 1 indicates the average performance of the current morphology in diverse environments, and the second term is a cost for each action taken by $\pi_m$ that modifies the morphology of the agent. $E$ is the set of environments used for evaluation.

**Environment evolution.** We model environment evolution as an MDP problem, denoted by the tuple $\{\mathcal{S}_e, \mathcal{C}, p_e, r_e, \rho\}$. The state $s_{\beta t}^e = (G_{\beta t}, \theta_{\beta t}^E)$ in the set of state space $\mathcal{S}_e$ includes the agent's topology $G_{\beta t} = (\mathcal{V}_{\beta t}, \mathcal{E}_{\beta t})$ and the parameter of environment $\theta_E \in \Theta_E$. The transition dynamics $p_e(s_{\beta t+1}^e|s_{\beta t}^e, c_{\beta t})$ reflects the changes in the state by taking an action at time-step $\beta t$. We define the reward for $\pi_e$ as the learning progress of multiple morphologies in the current environment. Thus, training $\pi_e$ will encourage $\pi_m$ to optimize the morphology of better generalization sooner. The reward $r_e$ at time-step $\beta t$ is denoted as

$$r_{\beta t}^E = \mathcal{P}_{\beta t} - \mathcal{P}_{\beta t-1}, \quad where \quad \mathcal{P}_{\beta t} = \mathcal{R}(H, \theta_{\beta t}^E, G_{\beta t}) - \mathcal{R}(H, \theta_{\beta t-1}^E, G_{\beta t}). \tag{2}$$

in Eq. 2, $\mathcal{P}_k$ is the learning progress of the RL agent in an environment defined on $\mathcal{R}(H, \theta_{\beta t}^E, G_{\beta t})$, which denotes the expected return of the RL agent $G_{\beta t}$ of $H$ time-steps evaluated in environment $\theta_k^E$.

**Short evaluation window.** Since $r_m$ and $r_e$ are respectively based on the training process of $\pi$ on different morphologies or environments, it is cost but necessary to rollout $\pi$ periodically to collect data. However, Hejna et al. (2021) demonstrates that evaluating with short horizon is enough to provide informative feedback to reflect the training process of the current policy. In light of this, we run a short evaluation window for $\pi$ after every period of environment steps taken by the agent. In each evaluation window, we rollout multiple morphologies of the best performance by $\pi$ in several most recent environments, and then we can calculate $r_m$ and $r_e$ based on the evaluation results. More details refer to Appendix. C.

In this paper, we use a standard policy gradient method, PPO (Schulman et al., 2017), to optimize these three policies.

## 4    ALGORITHM OF MECE

In MECE, we need jointly train three policies: control policy $\pi$ learns to complete tasks, morphology policy $\pi_m$ learns to evolve the agent's morphology to be more adaptive to diverse environments, and environment policy $\pi_e$ learns to modify the training environments more challenging. At first glance, training a single RL policies may appear more difficult than training a single RL policy, as it needs the collection of additional data via interaction. However, MECE enables three policies to help in each other's training through the co-evolving mechanism, where each policy is trained on a curriculum of easy-to-hard tasks utilizing dense feedback from the training experience of other policies. By

iterating this co-evolution process, MECE considerably improves the training efficiency of each policy, resulting in an RL agent of morphology that is more generalizable across environments.

In each episode, the control policy $\pi$ starts from training on an initial agent in a randomly sampled environment (refer to Line.3-9 in Alg. 1). This initial agent's skeleton structure is relatively simple and is easy to learn according to its naive morphology. Even though the naive agent is not generalized due to its constrained morphology, it can initially provide dense feedback for training $\pi_m$ and $\pi_e$. On the other hand, beginning from an agent with a random morphology is theoretically feasible but inefficient when initializing a complicated morphology that is difficult to learn to control and yields uninformative feedback for training the other policies.

We assume that the control policy is now mature on the current morphology after training. Then MECE proceeds to a series of co-evolution phases, where a phase is associated with achieving robustness across the evolved morphologies and environments (refer to Line.11 in Alg. 1). In each phase, as shown in Line.3-4 of Alg. 2, we first evaluate the control policy to collect rewards for training $\pi_m$ and $\pi_e$. Note that this step is not cost, as the RL agent executes environment steps with a short horizon. After that, we alternately apply $\pi_m$ and $\pi_e$ to co-evolve the agent's morphology and the training environments based on two criteria of $r_m$ and $r_e$ (refer to Line.5 and 9 in Alg. 2).

---

**Algorithm 1** MECE

1: **Initialize:** control policy $\pi$, morphology policy $\pi_m$, environment policy $\pi_e$, dataset $\mathcal{D} \leftarrow \emptyset$, initial agent morphology $m_0$, initial environment parameter $\theta_0^E$;
2: **while** Not reaching max iterations **do**
3:     **for** $t = 0, 1, \cdots, \tau_{\max}$ **do**    ▷ RL exploration
4:         Sample control action $a_t \sim \pi$;
5:         $s_{t+1} \leftarrow \mathcal{T}(s_{t+1}|s_t, a_t)$;
6:         $r_t \leftarrow$ environment reward;
7:         $m_{t+1} \leftarrow m_t, \theta_{t+1}^E \leftarrow \theta_t^E$;
8:         Store $(s_t, r_t, a_t, s_{t+1}, m_t, \theta_t^E)$ into $\mathcal{D}$;
9:     **end for**
10:    Update $\pi$ with PPO using samples in $\mathcal{D}$;
11:    $(m_{t+1}, \theta_{t+1}^E) = $ CO-EVO$(m_t, \pi, \theta_t^E)$;
12: **end while**

---

**Algorithm 2** CO-EVO

1: **Input:** Current morphology $m_{\alpha t}$, control policy $\pi$, training environment $\theta_{\beta t}^E$, dataset $\mathcal{D}$.
2: **Procedure** CO-EVO$(m_{\alpha t}, \pi, \theta_{\beta t}^E)$:
3: Evaluate $\pi$ with morphology $m_{\alpha t}$ in $\theta_{\beta t}^E$;
4: Calculate reward $r_m$ and $r_e$ following Eq. 1 and Eq. 2;
5: **if** $r_m \leq \delta_m$ **then**   ▷ Modify the agent's morphology
6:     Apply $\pi_m$ to modify $m_{\alpha t}$ to $m_{\alpha t+1}$;
7:     Update $\pi_m$ with PPO using samples in $\mathcal{D}$;
8: **else**
9:     **if** $r_e \leq \delta_e$ **then**  ▷ Modify the training environment
10:         Apply $\pi_e$ to modify $\theta_{\beta t}^E$ to $\theta_{\beta t+1}^E$;
11:         Update $\pi_e$ with PPO using samples in $\mathcal{D}$;
12:     **end if**
13: **end if**
14: **Return** $(m_{\alpha t+1}, \theta_{\beta t+1}^E)$

---

**Reward criteria.** As mentioned in section 3, $r_m$ and $r_e$ indicate the learning progress of the current morphology in the training environment respectively. In particular, a nominal value of $r_m$ corresponds to the adaptability of the current morphology to different environments and indicates that training with the current morphology cannot significantly increase $\pi$'s performance. In this instance, we should apply $\pi_m$ to optimize the morphology to increase its generalization to environments. Similarly, a minor $r_e$ indicates that modifying the agent's morphology will have a negligible effect on its performance in the current environment, and the environment should be modified to be more challenging to boost the morphology evolution. As a result, we propose two independent criteria for $r_m$ and $r_e$, which allows MECE to adapt to the learning progress of morphology and environment and apply $\pi_m$ or $\pi_e$ to produce corresponding evolution and alterations.

We now have an agent of newly evolved morphology or a modified training environment, and then forward to the next iteration of training the control policy $\pi$ on them. Using $r_m$ and $r_e$, MECE has achieved the alternating evolution of morphology and environment. MECE not only improves the robustness of the morphology to various environment, but also improves the learning efficiency of the policies through the use of dense feedback.

## 5 EXPERIMENTS

We design our experiments to answer the following questions: (1) Given dynamic environments, does our method outperform previous methods in terms of convergence speed and final performance? (2) Does our method create agents of better generalization to diverse environments?

## 5.1 Environments Setup

Diverse environments have vary features that require the agent to evolve out different morphologies, e.g., a bipedal agent has too short legs to go across a high obstacle, or a agent of four legs navigate on rough road more smoothly than one of one/two legs. In our experiments, we conduct three environments that requires various morphologies to complete the tasks. We use a tuple $\theta^E$ of environment parameters to denote the possible physical features of environments.

In experiments, we have three types of environments: **(1) 2d locomotion:** In the 2D locomotion environment, agents are limited to movement in the X-Z plane. The state $s_T$ is given by the final $(x, z)$ positions of the morphology joints. We evaluate morphologies on three tasks: running forwards, running backwards, and jumping. Environment policy $\pi_e$ takes actions to change the roughness of terrains, that is controlled by $\theta^E$. **(2) 3d locomotion:** where a 3D agent's goal is to move as fast as possible along x-axis and is rewarded by its forward speed along x-axis. Environments have a fixed number of obstacles and terrains of different roughness which are controlled by $\theta^E$. $\pi_e$ can not only change the roughness of terrains in the environments, also learns to modify the average distance between obstacles. **(3) Gap crosser:** Gap Crosser, where a 2D agent living in an $xz-$plane needs to cross periodic gaps and is rewarded by its forward speed. The width of the gap is controlled by $\pi_e$. In order to avoid unlearnable environments, the upper and lower limits of the gap width are limited. More details refer to Appendix. A.

**Baselines.** We compare our method with the following baselines that also optimize both the skeletal structure and joint attributes of an agent. (1)Neural Graph Evolution (NGE)(Wang et al., 2019b), which is an ES-based method that uses GNNs to enable weight sharing between an agent and its offspring. (2) Transform2Act (Yuan et al., 2022), that optimizes an RL policy to control the evolution of agents. Note that both baselines do not have diverse environments in the original code. For a fair comparison, we apply an environment set randomly sampled from a uniform distribution in the training and test phase for them.

## 5.2 Main results

In Fig. 2(a)-(c), we report the MECE and all baseline method performance on test environments for three environments. In terms of the learning efficiency and final performance, MECE outperforms baseline methods in all environments. The results show that while Transform2Act's final performance can be improved by periodically changing the training environments, MECE's learning efficiency is still notably higher than that of Transform2Act. The best morphologies found by each approach are listed in Fig. 2(d)-(f) for easier comparison. For 2d locomotion MECE develops a morphology that resembles a crawling animal that can move fast over terrain. In 3D locomotion, MECE has developed a spider-like framework. MECE evolves a more streamlined structure in comparison to Transform2Act so that it can more possibly avoid obstacles. Finally, the Hopper-like agents developed by Transform2Act and MECE are comparable in the Gap crosser that can jump across gaps. Overall, MECE-optimized morphologies have superior structural symmetry and are more consistent with the characteristics of biological evolution in various environments.

## 5.3 Ablation Study and Analysis

We product out a series of ablation studies to prove the effectiveness of each part in MECE. We list the introduction information of each ablation study in Tab. 1, and report the results in Fig. 3. Note that in each ablation study, we only change one part of MECE and maintain the other parts the same as the original MECE, and all results are the test performance evaluated on the same environment set in Fig. 2 (b).

**Ablation Study I.** In this ablation study, we focus on the effectiveness of $\pi_e$ for the training and report the results in Fig. 2(a). Note that the initial environment is relatively simple, while the final environment that evolved by $\pi_m$ is more challenging. The results show that the evolved morphology and control policy trained in the diverse environments are more general than trained in the fixed environment, no matter the environment is simple or not. On the other hand, compare the performance of MECE with MECE (periodic envs), we can find that $\pi_e$ helps the training in terms of the efficiency and the final performance. This is because training in the environment modified by $\pi_e$, compared to

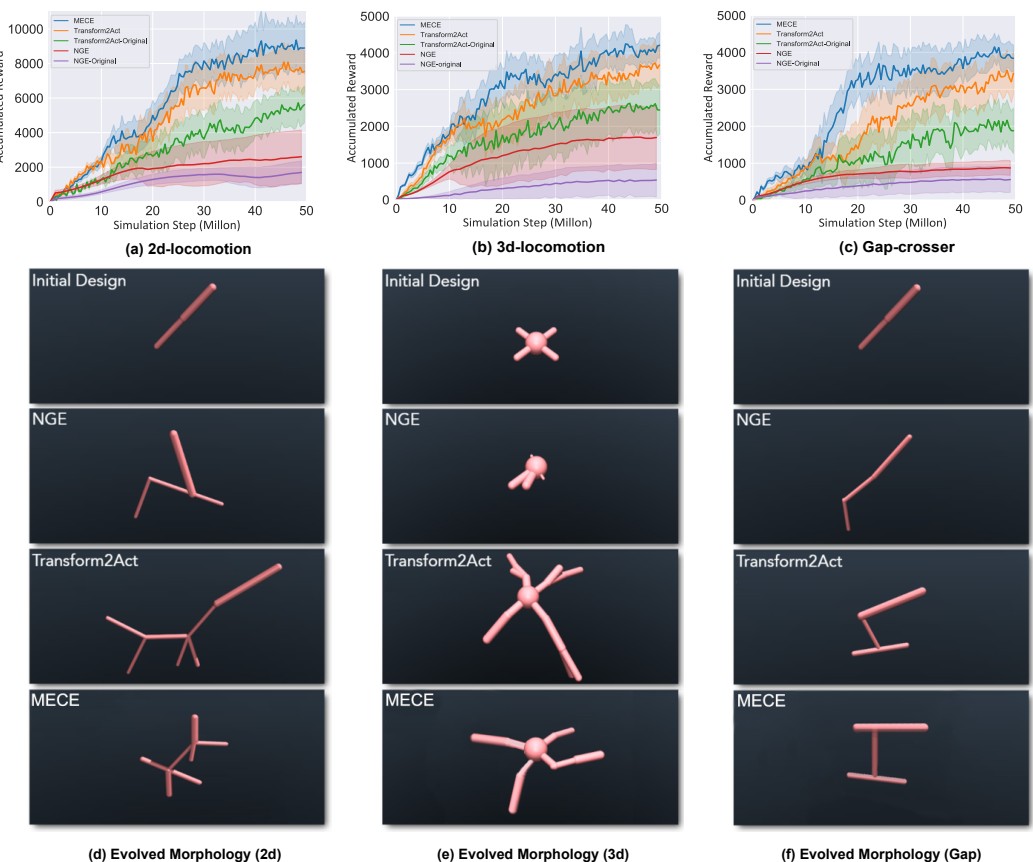

Figure 2: **Baselines comparison and optimized morphology.** In Fig.(a)-(c), we report the evaluation results of MECE and baseline methods in three environments, and we plot the accumulated rewards (mean± std averaged over 6 random seeds) against the number of simulation steps for all methods. For fair comparison, we add periodically changing environments that randomly sampled from a fixed distribution in the training of baseline methods. MECE has better learning efficiency and final performance than all baseline methods. In Fig.(d)-(f), we list the optimal morphology that evolved by each method. Intuitively, the structural symmetry and environmental adaptability of the MECE-optimized morphology is better. Especially in 3d locomotion, the agent developed by MECE is better at navigating terrain and avoiding obstacles.

| Ablation Study | Definition of Legends |
|---|---|
| Ablation study-I for $\pi_e$. Fig. 3(a) | **original**: Periodic environments generated by $\pi_e$.
**periodic envs-random**: Periodic environments of randomly sampled.
**fixed envs-initial**: Initial and easy environment.
**fixed envs-final**: Fixed environment generated by $\pi_e$. |
| Ablation study-II for $\pi_m$. Fig. 3(b) | **original**: Dynamic morphology evolved by $\pi_m$.
**random morph**: Randomly mutate the morphology.
**fixed morph-initial**: Fixed agent of the initial morphology.
**fixed morph-final**: Fixed agent of the final morphology evolved by $\pi_m$. |
| Ablation study-III for dynamic update window. Fig. 3(c) | **Original**: Dynamic evaluation window.
**fixed update window**: Update $\pi_e$ and $\pi_m$ at a fixed frequency. |
| Ablation study-IV for $r_m$ and $r_e$. Fig. 3(d) | **original**: Follow the reward setting of $\pi_m$ ($\pi_e$) in Eq. 1 (Eq. 2).
**reward-i**: Remove the reward for $\pi_m$.
**reward-ii**: change $r_m$ to Eq. 1.
**reward-iii**: change $r_e$ to Eq. 2. |

Table 1: List of ablation studies. This list includes definitions for each set of controls as well as the objectives of ablation study I-IV. Since we only validated one design of MECE in each ablation study, only one design from each control group was different from MECE (original).

Figure 3: **Ablation study.** We design four ablation studies to illustrate the advantages of each part in MECE. The results of ablation study I and II show the effectiveness of $\pi_e$ and $\pi_m$ on the learning efficiency and generalization. Ablation study III proves that applying $\pi_e$ and $\pi_m$ adaptive to the training process of the control policy can improve its robustness. In ablation study IV, we try different setting of the reward function of $\pi_e$ and $\pi_m$, and the results prove that the original reward setting is optimal. More details of each ablation study can be found in Tab. 1.

the randomly generated one, avoids the learning gap between two extremely different environments, and is smoother for $\pi$ and $\pi_m$ to learn. On the other hand, $\pi_e$ learns to generate environment to accelerate the learning progress of $\pi$ and $\pi_m$, which is shown clearly in the learning curves of 10 - 20 million simulation steps in Fig. 3(a).

**Ablation Study II.** The purpose of this ablation study is to demonstrate how $\pi_m$ can produce morphology that is more adaptable to diverse environments and increase learning effectiveness. The results are shown in Fig. 3(b). When comparing the learning curves for MECE (random morph) and MECE (original), the former has a far higher early learning efficiency than the latter. This is due to $\pi_e$ remaining constant and the environment not changing significantly, which prevents the morphology from being adequately adapted to a variety of habitats, even some comparable environments. Theoretically, the "fixed morph-final" morphology should be able to be more adaptive to environments and, with training, reach competitive performance. Although it performs better than "random morph" and "fixed morph-initial," "MECE(original)" is still far behind. This is because that the final morphology is more complicated and therefore directly teaching its control policy to adapt to various environments would be more challenging.

**Ablation Study III.** Through a co-evolutionary process of adaptive criterion on $r_m$ and $r_e$, $\pi_m$ and $\pi_e$ learn to evolve the morphology and environment. To determine whether this design is effective, we perform an ablation experiment. The results are shown in Fig 3(c). For "MECE (fixed evaluation window)", every fixed number of environment steps, $\pi_m$ and $\pi_e$ take actions to change the agent's morphology and environment. The results indicate that by removing the dynamic window, MECE's learning effectiveness has drastically decreased. It is challenging to coordinate the training developments of the three policies, particularly the control policy, without a dynamic evaluation window. Only a somewhat steady control policy, as was already mentioned, can give the other two reliable feedback for training. Moreover, the performance of Transform2Act is taken into account for easy comparing. Even though "MECE(fixed evaluation window)" is less effective, a competitive final performance to Transform2Act is still feasible.

**Ablation Study IV.** We designed this ablation investigation to verify the reward of $\pi_m$ and $\pi_e$ in MECE, and the findings are displayed in Fig. 3(d). Note that just one incentive is altered in each scenario in this experiment; all other rewards remain unchanged. We explore two scenarios for $\pi_m$: the first involves removing the reward ("MECE (reward-i)"), same to the setting of transform2Act), and the second involves substituting the formal of $r_e$ ("MECE (reward-ii)"), i.e., accelerating learning progress. We take into account employing learning progress (in the form of $r_m$, "MECE (reward-iii)") for $\pi_e$. The results show that the original design has obvious advantages in both learning efficiency and final performance. For the issues addressed by MECE, $\pi_m$ can be thought of as the meta learner of $\pi$, whereas $\pi_e$ is the meta learner of the latte. In order to help $\pi$ perform better on various tasks, $\pi_m$ learns to modify the agent's morphology during training, and $\pi_e$ speeds up both of their learning processes. Therefore, in both theoretical and practical tests, the original design is the most logical.

## 5.4 CASE STUDY: CO-EVOLUTION BETWEEN MORPHOLOGY AND ENVIRONMENT

To further comprehend the co-evolution of morphology and environment in MECE, we perform a case study in 2d locomotion and present results in Fig. 4. The y-axis of the point plot indicates the ratio of the training environment's roughness to the current morphology's height. Fig. 4(a)-(f) are schematic descriptions of the current morphology and the training environment corresponding to it.

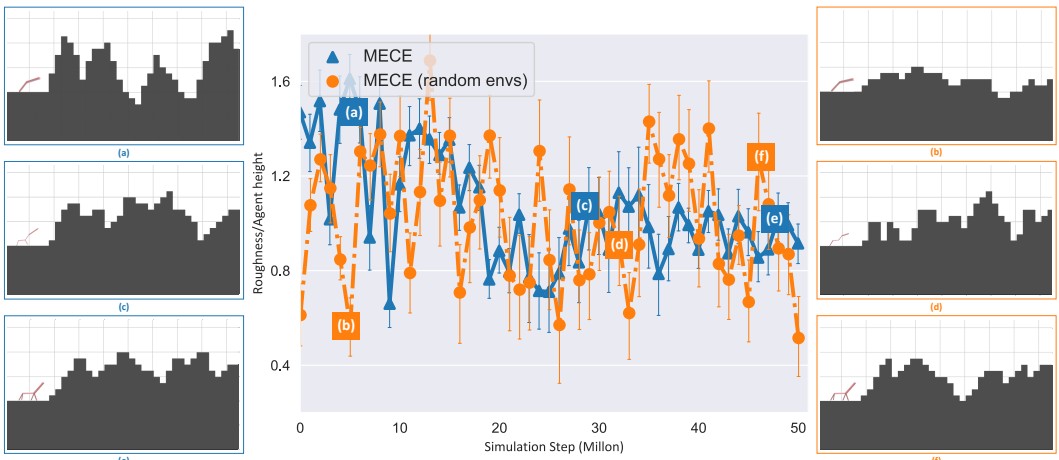

Figure 4: **Case study results.** This figure more vividly illustrates a schematic diagram of the co-evolution of morphology and environment during training. The ordinate in the illustration shows the proportion of the environment's roughness to the agent's height. Since the deviation is minimal and the roughness of the training environment produced by MECE is about similar to the agent's height (the ratio is near to 1), it can be observed from the pointplot that $\pi_e$ can not only guarantee challenging environments but also more stable than randomly sampled. Fig.(a)-(f) compare the effectiveness of $\pi_e$ that correspond to changes in the training environments (blue indicates original MECE, and orange indicates random environments). Contrarily, it is evident that no $\pi_e$ is likely to result in exceptionally difficult environments (Fig.(d)), while the environment produced by $\pi_e$ is challenging but learnable (Fig.(c)).

Generally, an effective training environment should meet two conditions simultaneously. The first is learnable, and the RL agent can collect training data via exploration. The second factor is the level of difficulty; environments that are either too easy or too challenging may reduce the learning efficiency of the RL agent. Therefore, in 2D locomotion, a reasonable environment is one in which the environment's roughness is quite large (challenging), but not so large as to render the agent unconquerable (learnable) during exploration. Using the results of the pointplot, the environment policy can ensure that the ratio of environmental unevenness during the training process to the height of the current agent is approximately 1, which satisfies the training environment criteria. At contrast, despite the fact that the randomly generated environment can ensure the ideal in some phases, it is extremely unstable (the standard deviation is visibly high), which would undoubtedly lower the learning effectiveness.

Specifically, comparing Fig. 4(a) and (c), the environment generated by $\pi_e$ in the early stage of training is unlearnable for the current agent, whereas in the middle stage of training, the training environment is relatively flat, but challenging, especially in terms of the number of occurrences of a particular feature. The roughness of the environment on the right fluctuates frequently. Note again compared to Fig. 4(d) that the right side of the randomly produced environment is unlearnable for the current agent in the middle of training. This phenomena does not occur in MECE because $\pi_e$ modifies the environment to be morphology-adaptive. Comparing Figs. 4(e) and (f), the environment formed by $\pi_e$'s slope exhibits noticeable but relatively subtle alterations. This is due to the fact that the existing morphology has a high degree of adaptation to varied contexts, and environments with more extreme alterations have less effect on the morphology's optimization. To better train the control policy, an environment of moderate difficulty should be adopted.

## 6 CONCLUSION

In this paper, we offer a framework for morphology-environment co-evolution that evolves the agent's morphology and the training environment in alternation. Compared to previous morphology optimization approaches, ours is more sample-efficient and learns to develop morphologies that are more robust across diverse environments. Experiments demonstrate that our approach outperforms baseline methods in terms of convergence speed and overall performance. In the future, we are interested in using curriculum RL approaches to further improve the learning efficiency of our approach so that it can adapt to more challenging environments and tasks.

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

# A    Environment Details

In this part, we share further information about the three experiment environments. A simple skeleton structure is used to initialize the agent. Each joint of the agent is connected to its parent joint via a hinge. The environment state of each joint contains its joint angle and velocity. In addition to the height, phase, and global velocity, we also include additional information for the root joint, such as the phase and height. Zero padding is employed to ensure that the length of each joint state is same. The attribute characteristic of each joint includes the bone vector, bone size, and motor gear value. Using a loosely-defined attribute range, each attribute is normalized within the interval $[-1, 1]$.

**2d locomotion**    The agent in this environment lives inside an $xz$-plane with a terrain ground. Each joint of the agent is allowed to have a maximum of three child joints. For the root joint, we add its height and 2D world velocity to the environment state. The enviornment reward function is defined as:

$$r_t = |p_{t+1}^x - p_t^x|/\delta t + 1, \tag{3}$$

where $p_t^x$ denotes the x-position of the agent and $\delta t = 0.008$ is the time step. An alive bonus of 1 is also used inside the reward. The episode is terminated when the root height is below 1.4.

**3d locomotion**    The agent in this environment lives freely in a 3D world with an uneven ground. There are a fixed number of obstacles randomly scattered on the surface. Each joint of the agent is allowed to have a maximum of three child joints except for the root. For the root joint, we add its height and 3D world velocity to the environment state. The reward function is defined as:

$$r_t = |p_{t+1}^x - p_t^x|/\delta t - \omega \cdot \frac{1}{J} \sum_{u \in V_t} \|a_{u,t}\|^2, \tag{4}$$

where $\omega = 0.0001$ is a weighting factor for the control penalty term. $u$ denotes the each node of the skeleton structure $V_t$ of the agent at time-step $t$. $J$ is the total number of agent's joints, and the time step $\delta t = 0.04$.

**Gap crosser**    The agent in this environment lives inside an $xz$-plane. The terrain of this environment includes a periodic gap. The height of the initial terrain is at 0.2. The agent must cross these gaps to move forward. Each joint of the agent is allowed to have a maximum of three child joints. For the root joint, we add its height, 2D world velocity, and a phase variable encoding the periodic $x$-position of the agent to the environment state. The reward function is defined as:

$$r_t = |p_{t+1}^x - p_t^x|/\delta t + 0.1, \tag{5}$$

where the time step $\delta t = 0.08$. An alive bonus of 0.1 is used inside the reward. The episode is terminated with the root height is below 1.5.

**Implementation of morphology poicy $\pi_m$**    MuJoCo agents are specified using XML strings, during the morphology modify phase, we represent each agent's skeleton structure as an XML string and modify its content based on the morphology actions. At the start of the RL exploration stage, the modified XML string is used to reset the MuJoCo simulator and load the agent of newly-modified morphology.

**Implementation of environment policy $\pi_e$**    Environment policy modify the training environment by taking actions to change the environment parameters. For locomotion environments, we sample terrain height maps using random Gaussian mixtures. There is an environment tuple of two parameters to control the generation of terrains. The first environment parameter is the maximum height of the terrain, which is limited to $2.4(2.7)$ in 2d (3d) locomotion. The second environment parameter is to control the variance of the sampled environments, which is restricted in $[2.4, 7.2]$ for 2d, and $[2.7, 5.4]$ for 3d. In 3d locomotion, we have one more environment parameter in the tuple to control the averaging spacing between the obstacles, which is restricted in $[1.6, 4.4]$.

## B    ADDITIONAL EXPERIMENTS RESULTS

To respond the request of reviewer QUPT, we report the changing environments and agent's morphology during the training in Fig. 5. The results show that when the evolved morphology is relatively complex, the training environment focuses more on complex cases than the randomly sampled training environments with environment generator. On the other hand, the prevalence of easy environments significantly decreases as morphology evolves (indicating a higher ability to move in different environments).

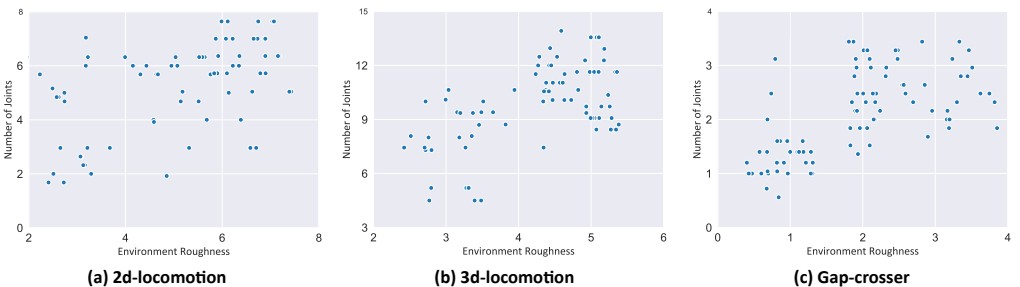

**(a) 2d-locomotion**       **(b) 3d-locomotion**       **(c) Gap-crosser**

Figure 5: This figure shows the changing of the agent's morphology and environments in the training phase. The results illustrate that with the environment generator ($\pi_e$), the training environment focuses more on complex cases (higher roughness). The presence of agents with more complex morphologies is more noticeable among them. Thus, the environment generator creates the training environment based on how the morphology changes during training.

## C    SHORT EVALUATION WINDOW (SEW) IN MECE

In Alg. 2, we modify the agent's morphology and the training environment based on the control policy's learning progress. In light of the previous method (Hejna et al., 2021), we construct a short evaluation window (SEW) on $\pi$ to collect data for computing $r_m$ and $r_e$. In accordance with Sec. 3, we should evaluate the present morphology in many settings for $r_m$ in Eq. 1 and several morphologies in the current training environment for $r_e$ in Eq. 2. In this short evaluation phase, we therefore maintain some of the most recent morphologies and training environments. We attempted to save several combinations of morphology and number of contexts for this purpose. All experimental and control data are provided under the assumption that the optimal combination exists. We list all tried combination in Tab. 2.

## D    HYPERPARAMETERS AND TRAINING PROCEDURES

In this section, we present the hyperparameters searched and used for MECE in Tab. 2 and the hyperparameters for baseline in Tab. 3 and Tab. 4. All models are implements by PyTorch (Paszke et al., 2019). For control policy and morphology policy, we use the PyTorch Geometric package (Fey & Lenssen, 2019) and GraphConv (Morris et al., 2019) as the GNN layer. When training the policy with PPO, we adopt generalized advantage estimation (GAE) (Schulman et al., 2016). For the baselines, we employ the released code of NGE and Transform2Act to build our version.

For fair comparison, we employ the same GNN architecture for MECE and baselines. In addition, we ensure that the number of simulation steps used for optimization is the same for both the baseline methods and ours. MECE and Transform2Act optimizes the policy with a batch size of 50000 for 1000 epochs, totaling 50 million simulation steps. NGE employs a population of 20 agents, and each agent is trained with a batch size of 20,000 for 125 generations, totaling 50 million simulation steps.

| Hyperparameter | Values Searched & Selected |
|---|---|
| Policy GNN Layer Type | GraphConv |
| Number of morphologies in SEW | $2, \mathbf{3}, 4$ |
| Number of environments in SEW | $2, 3, \mathbf{4}, 5$ |
| GNN Size ($\pi_m$) | $(32, 32, 32), (64, 64, 64), (128, 128, 128), (\mathbf{256}, \mathbf{256}, \mathbf{256})$ |
| GNN Size ($\pi$) | $(32, 32, 32), (\mathbf{64}, \mathbf{64}, \mathbf{64}), (128, 128, 128), (256, 256, 256)$ |
| Hidden layers for $\pi_e$ network | $\mathbf{2}, 3, 4$ |
| Hidden units for $\pi_e$ network | $\mathbf{200}, 300, 400$ |
| Policy Learning Rate ($\pi$) | $\mathbf{5e-5}, 1e-4, 3e-4$ |
| Policy Learning Rate ($\pi_m$) | $\mathbf{5e-5}, 1e-4, 3e-4$ |
| Policy Learning Rate ($\pi_e$) | $\mathbf{3e-4}, 5e-4, 1e-4$ |
| Value GNN Layer Type | **GraphConv** |
| Value Activation Function | Tanh |
| Value GNN Size | $(\mathbf{64}, \mathbf{64}, \mathbf{64}), (128, 128, 128), (256, 256, 256)$ |
| Value MLP Size | $(256, 256), (\mathbf{512}, \mathbf{256}), (256, 256, 256)$ |
| Value Learning Rate | $1e-4, \mathbf{3e-4}$ |
| PPO clip $\epsilon$ Pize | **0.2** |
| PPO Batch Size | $10000, 20000, \mathbf{50000}$ |
| PPO Minibatch Size | $512, \mathbf{2048}$ |
| Num. of PPO Iterations Per Batch | $1, 5, \mathbf{10}$ |
| Num. of Training Epochs | **1000** |
| Discount factor $\gamma$ | **0.95** |
| GAE $\lambda$ | $\mathbf{0.99}, 0.995, 0.997, 0.999$ |

Table 2: Hyperparameters searched and used by MECE. The bold numbers among multiple values are the final selected ones.

| Hyperparameter | Values Searched & Selected |
|---|---|
| Num. of Skeleton Transforms $N_{\mathrm{s}}$ | $3, \mathbf{5}, 10$ |
| Num. of Attribute Transforms $N_{\mathrm{z}}$ | $\mathbf{1}, 3, 5$ |
| Policy GNN Layer Type | GraphConv |
| JSMLP Activation Function | Tanh |
| GNN Size (Skeleton Transform) | $(64, 64, 64), (\mathbf{128}, \mathbf{128}, \mathbf{128}), (256, 256, 256)$ |
| JSMLP Size (Skeleton Transform) | $(\mathbf{256}, \mathbf{256}), (512, 256), (256, 256, 256)$ |
| GNN Size (Attribute Transform) | $(64, 64, 64), (\mathbf{128}, \mathbf{128}, \mathbf{128}), (256, 256, 256)$ |
| JSMLP Size (Attribute Transform) | $(\mathbf{256}, \mathbf{256}), (512, 256), (256, 256, 256)$ |
| GNN Size (Execution) | $(\mathbf{64}, \mathbf{64}, \mathbf{64}), (128, 128, 128), (256, 256, 256)$ |
| JSMLP Size (Execution) | $(\mathbf{128}, \mathbf{128}), (256, 256), (512, 256), (256, 256, 256)$ |
| Diagonal Values of $\Sigma^{\mathrm{z}}$ | $1.0, 0.04, \mathbf{0.01}$ |
| Diagonal Values of $\Sigma^{\mathrm{e}}$ | $\mathbf{1.0}, 0.04, 0.01$ |
| Policy Learning Rate | $\mathbf{5e-5}$ |
| Value GNN Layer Type | **GraphConv** |
| Value Activation Function | Tanh |
| Value GNN Size | $(\mathbf{64}, \mathbf{64}, \mathbf{64}), (128, 128, 128), (256, 256, 256)$ |
| Value MLP Size | $(256, 256), (\mathbf{512}, \mathbf{256}), (256, 256, 256)$ |
| Value Learning Rate | $1e-4, \mathbf{3e-4}$ |
| PPO clip $\epsilon$ Pize | **0.2** |
| PPO Batch Size | **50000** |
| PPO Minibatch Size | **2048** |
| Num. of PPO Iterations Per Batch | $1, 5, \mathbf{10}$ |
| Num. of Training Epochs | **1000** |
| Discount factor $\gamma$ | **0.95** |
| GAE $\lambda$ | **0.995** |

Table 3: Hyperparameters searched and used by Transform2Act in our version. The bold numbers among multiple values are the final selected ones.

| Hyperparameter | Values Searched & Selected |
|---|---|
| Num. of Generations | **125** |
| Agent Population Size | $10, \mathbf{20}, 50, 100$ |
| Elimination Rate | $\mathbf{0.15}, 0.2, 0.3, 0.4$ |
| GNN Layer Type | GraphConv |
| MLP Activation | Tanh |
| Policy GNN Size | $(32, 32, 32), (\mathbf{64}, \mathbf{64}, \mathbf{64}), (128, 128, 128)$ |
| Policy MLP Size | $(\mathbf{128}, \mathbf{128}), (256, 256), (512, 256)$ |
| Policy Log Standard Deviation | $\mathbf{0.0}, -1.6$ |
| Policy Learning Rate | $\mathbf{5e-5}$ |
| Value GNN Size | $(32, 32, 32), (\mathbf{64}, \mathbf{64}, \mathbf{64}), (128, 128, 128)$ |
| Value MLP Size | $(128, 128), (256, 256), (\mathbf{512}, \mathbf{256})$ |
| Value Learning Rate | $\mathbf{3e-4}$ |
| PPO clip $\epsilon$ | **0.2** |
| PPO Batch Size | $\mathbf{20000}, 50000$ |
| PPO Minibatch Size | **2048** |
| Num. of PPO Iterations Per Batch | **10** |
| Discount factor $\gamma$ | $0.99, \mathbf{0.995}$ |
| GAE $\lambda$ | **0.95** |

Table 4: Hyperparameters searched and used by NGE in our version. The bold numbers among multiple values are the final selected ones.

