# OpenReview forum: "Curriculum Reinforcement Learning via Morphology-Environment Co-Evolution"
_ICLR.cc/2023/Conference — Submitted to ICLR 2023_

### Official Review · Reviewer_QUPT · 2022-10-24

**Confidence:** 4
**Correctness:** 3
**Technical Novelty And Significance:** 3
**Empirical Novelty And Significance:** 3
**Recommendation:** 5

**Clarity, Quality, Novelty And Reproducibility:**

The paper is clear and generally easy to follow, although the clarity can be improved with minor changes (indicated above). The technical quality of the work is high. The idea of the joint curriculum learning over the environment and agent morphology is novel, as far as I’m concerned. The authors provided the hyperparameter values for their experiments (but not the code itself).

**Strength And Weaknesses:**

## Strengths

- The paper takes on an interesting problem in deep RL, jointing learning an environment and morphology of the agent for zero-shot generalization on previously unseen tasks.
- This paper provides a comprehensive set of experiments and, particularly, ablation studies to make sure that all aspects of the MECE are important and necessary for superior performance.

## Weaknesses

- I have some concerns and questions regarding the morphology-environment co-evolution in MECE.
    - It is unclear to me why the authors update $\pi_m$ and $\pi_e$ with samples from D before using those to modify either the morphology or the environment. I think the opposite makes more sense.
    - It is unclear to me why it is not possible to modify both (they are in the different blocks of the if-else statement). Have the authors tried ablating this part of the algorithm?
    - Most importantly, I’m concerned about the new hyperparameters $\delta_m$ and $\delta_e$. It seems that these change the nature of the CO-EVO significantly, but fine-tuning these would be difficult because there are no intuitive values that would work for any environment. I’d like to hear what the authors have to say about this.
- Rewards for each environment used in these papers are different (Appendix A). I suspect this limits the generality of the method and makes the “off-the-shelf” usage of it impossible. Instead, it seems, that people would need to spend a lot of time and compute to find a good reward signal for the environment.
- I would expect this paper to include graphs showing how the morphology and environment change for all the domains used in experimentation. Specifically, plotting morphology metrics (number of joints, legs) and environment (e.g. roughness) as a y-axis throughout the training. I would expect this to be in the paper for all three domains.
- It is unclear to me if the AddJoint action has any arguments or just adds a joint at a random leg of the agent. Have the authors considered a more parametrized morphology-changing interface?

### Minor issues.

- Please increase the font size of the legend, etc in all figures. It is very difficult to read these.
- Please bring Table 1 forward, near the beginning of Section 5.3.
- I think in the second paragraph of Section 5.3 (first sentence) you should refer to Figure 3, not 2.
- It is common to have the pseudo-code within a Procedure indented. (Algorithm 2)
- “Unlike” is capitalized unnecessarily on Page 2, in the paragraph that starts with “The interplay …”
- E is undefined in Eq (1)
- In the first sentence of Section 4, please use $\pi$ and $\pi_m$ rather than ***pi*** and ***pim.***
- The second sentence in Section 4 suggests that “training three policies may be trivial than a single RL policy”. Is this a mistake (you meant the opposite)?
- Figure 2 doesn’t have (d) (e) and (f) labels as subfigures.
- The word “unconqueerable” needs to be fixed on page 9.

### Questions

- Is the test set used for evaluation in-distribution or out-of-distribution compared to training environment variations?
- What do the “Number of environments in SEW” and “Number of morphologies in SEW” specifically mean in Table 2?
- Were the morphologies of the agents and environment parameters constrained in any way?

**Summary Of The Paper:**

This paper proposes a curriculum learning approach for reinforcement learning (RL) that changes the morphology of the agent as well as the environment for training agents that are robust to test-time environments. The authors provide thorough experimental results and ablation studies in three domains and demonstrate the effectiveness of their method.

**Summary Of The Review:**

The idea behind the paper is interesting, and the authors provided thorough experiments in support of their method. That said, I still have some concerns regarding the applicability/generality of the method, lack of analysis on the exact curriculum induced by MECE, etc. I have added several questions which I hope the authors will respond to, as well as fix minor issues that I have identified.

---

> ### Author Response · Authors · 2022-11-18
> **Response to reviewer QUPT (2/2)**
>
> ### Q7: Is the test set used for evaluation in-distribution or out-of-distribution compared to the training environment?
>
> - The environments in the test set are randomly drawn from a uniform distribution, while the training environments are generated by $\pi_e$ as a curriculum in order to improve the optimization of morphology and RL policy for the uniform distribution.
> - We show that our curriculum learning approach leads to a morphology and RL policy that can generalize much better to the test set environments than that trained on random environments drawn from the uniform distribution. Please refer to Fig. 3 (a) for a comparison between MECE (original) and MECE (periodic evans - random).
> ### Q8: “Number of environments in SEW” and “Number of morphologies in SET” in Tab.2.
>
> - In order to compute $r_m$ and $r_e$ in Eq.1 and Eq.2, we need to rollout the control policy $\pi$ in several different environments or with different morphologies respectively to acquire $r_m$ and $r_e$. Hence, these two Numbers refer to the number of recent environments and the number of recent morphologies respectively and we maintain them in memory for a short evaluation window in each round.
>
> ### Q9: Were the morphologies of the agents and environment parameters constrained in any way?
> - Yes. We provided the details in Appendix.A.

---

> ### Author Response · Authors · 2022-11-18
> **Response to reviewer QUPT (1/2)**
>
> Thank you for your suggestions and comments! We have carefully corrected typos and removed ambiguities in the new version. We address your main concerns in the following:
>
> ### Q1: Why not apply $\pi_m$ ($\pi_e$) to modify the morphology (environment) and then update?
> - Perhaps there is a typo in your question, since that's what we did. Do you mean to ask why we update $\pi_e$ ($\pi_m$) before using it to modify the environment (morphology)? The reason is that we have no trajectory or transitions of $\pi_e$ ($\pi_m$) collected to compare different morphologies (environments) before the first update. So we have to apply them first before updating them. In fact, since $\pi, \pi_e, \pi_m$ are iteratively updated in each episode and we repeat such alternating updates, their order for rollout-update has little effect on the results.
>
> ### Q2: Why not modify both the environment and morphology at the same time?
> - Only changing one (the environment or morphology) at a time while fixing the other is critical for computing the reward $r_m$ in Eq. 1, which measures the learning progress of the control policy $\pi$ caused by the most recent morphology evolution. Hence, we need to rule out the impact of simultaneous changes in the environment.
> - Similarly, in order to accurately estimate $r_e$, we need to rule out the possible credits caused by changing the morphology, so we keep the morphology fixed.
> If we change both the environment and the morphology at the same time, it is difficult to tell whether the observed learning progress is due to the evolved morphology or the changes in the training environments.
>
> ### Q3: Hyperparameter $\delta_e$ and $\delta_m$
> The main reasons for having two hyperparameters are: (1) to obtain more accurate estimations of the rewards, $r_m$ and $r_e; (2) to reduce the cost of evaluating the RL policy every iteration; and (3) to obtain more informative training rewards. In particular, not having the two criteria may lead to the following problems:
>   - **Inaccurate rewards due to convergence instability**: Since rewards $r_m$ and $r_e$ are based on the evaluation of the control policy $\pi$, it is important to ensure the attribution is accurate, i.e., $r_m$ is only a result of morphology evolution and $r_e$ is only a result of environment modification. However, these rewards are learning progress based so they also heavily depend on the convergence of $\pi$, e.g., earlier stages may naturally has more progress than the later stages.
>   - A frequently changing environment or morphology will slow or even prevent $pi$ convergence, resulting in inaccurate $r_m$ and $r_e$ rewards that are primarily influenced by the convergence instability.
>   - **Costly on RL policy evaluation**: If we apply $\pi_e$ and $\pi_m$ more frequently, e.g., every iteration, the RL policy will take longer time to converge because the environment and morphology change in every iteration.
>   - **Rewards are less informative if too small**: the rewards $r_m$ and $r_e$ are not very informative to the training if their values are too small, indicates that there is little progress.
>   - In our experiments, we chose $\delta_e$ and $\delta_m$ based on the statistics of $r_m$ and $r_e$, as well as the interval in which their values primarily lay. This strategy can also be extended to other domains.
>
> ### Q4: Rewards for each environment used in these papers are different.
> - We follow the commonly used environment rewards for different domains defined in Mujoco. We have not added any extra reward for training $\pi_m$ or $\pi_e$.
>
> ### Q5: AddJoint actsion has any arguments or just adds a joint at a random leg of the agent.
> - In experimental settings of  this paper, $\pi_m$ can only add/delete one joint in each of its action. We agree that a more parametrized morphology-changing interface sounds more general, and we will further investigate it in future work, considering this is the first work coupling environment changing with morphology evolution in training a generalizable RL agent.
>
> ### Q6: Plotting morphology metrics (number of joints, legs) and environment (e.g. roughness) as a y-axis throughout the training.
> - We have plotted the changing of morphology metrics and environments in the new version (refer to Appendix.B). It shows a trend that the number of joints tends to increase with the increase of the environment's roughness.

---

> > ### Comment · Reviewer_QUPT · 2022-11-23
> > **Response to rebuttal**
> >
> > I thank the authors for the response.
> >
> > On Q1, it is unclear to me why is $\pi_m$ updated using samples in $\mathcal{D}$ if $\mathcal{D}$ used a different morphology. Wouldn't it be better to get new samples with modified morphology and use this up-to-date $\mathcal{D}'$ to update $\pi_m$ (same for $\pi_e$).
> >
> > Thank you for adding additional results in Appendix B. It would be great to know how number of joints and environment roughness change over time during the training. Figure 5 doesn't show that unfortunately.

---

> > > ### Author Response · Authors · 2022-11-25
> > > **Response to reviewer QUPT**
> > >
> > > Thank you for the response. We are glad to know that some of your concerns have been addressed. We provide a more detailed reply to your latest comments in the following.
> > > > It is unclear to me why is $\pi_m$ updated using samples in $\mathcal D$ if $\mathcal D$ used a different morphology.
> > >
> > > - The trajectory in $\mathcal D$ used to train the morphology policy $\pi_m$ is a trajectory of varying morphology (instead of a control trajectory under a fixed morphology) from the very beginning of the curriculum and every step records a change applied to the morphology. Hence, in our algorithm, the memory $\mathcal D$ in the $\it t-th$ iteration contains the trajectory of historic morphologies $\{ m_1, m_2, …, m_t \}$. The only difference between an up-to-date $\mathcal D’$ and the current $\mathcal D$ is that the former includes the latest morphology $m_{t+1}$, i.e., $\mathcal D’$ contains $ \{ m_1, m_2, …, m_t, m_{t+1}\}$. Since $m_{t+1}$ will be used in the next iteration to update $\pi_m$, this minor difference does not noticeably affect the final results as long as the learning converges.
> > >
> > > > It would be great to know how the number of joints and environment roughness change over time during the training.
> > >
> > > - Fig. 4 in the original paper provides a concrete example of how the environment and morphology changes throughout the training as a co-evolution curriculum. It shows that the environment (roughness) and morphology (agent’s height) will converge to an equilibrium.
> > > - Although Fig. 5 in Appendix B does not explicitly plot the time axis, the points from left to right are roughly ordered by time (from early to late) so they do reflect the trend of how the number of joints and environment roughness change over time, i.e., both start from small values (easier and simpler) and increases over time until reaching an equilibrium between them. In order to provide further evidence, we report the number of joints and the roughness (averaged over six different random seeds) at six different simulation steps from early to late stages in the table below. The result verifies the trend (implicitly) shown in Fig. 5.
> > > - We will add a plot for the numbers in this table when uploading a new version of the paper is allowed.
> > >
> > > |Simulation Steps (Million)|Joints of agent (2d)|Environment Roughness (2d)|
> > > |:--|:--|:--|
> > > | 5 | 3.32 | 2.24 |
> > > | 10 | 3.86 | 3.82 |
> > > | 20 | 6.17 | 4.32 |
> > > | 30 | 6.33 | 5.04 |
> > > | 40 | 6.64 | 6.33 |
> > > | 50 | 6.48 | 6.73 |
> > >
> > > |Simulation Steps (Million)|Joints of agent (3d)| Environment Roughness (3d)|
> > > |:--|:--|:--|
> > > | 5 | 6.32 | 2.71 |
> > > | 10 | 7.16 | 2.93|
> > > | 20 | 9.17 | 3.24 |
> > > | 30 |11.48 | 4.55 |
> > > | 40 | 10.84 | 4.93 |
> > > | 50 | 10.49 | 5.07 |
> > >
> > > |Simulation Steps (Million)|Joints of agent (Gap)|Environment Roughness (Gap)|
> > > |:--|:--|:--|
> > > |5| 1.17 | 0.44 |
> > > |10 | 1.17 |1.33|
> > > | 20 |1.64 |1.93|
> > > | 30 |2.33|2.44|
> > > | 40 |2.85 |3.32|
> > > | 50 |2.85| 3.76|

---

### Official Review · Reviewer_ugJV · 2022-10-25

**Confidence:** 4
**Correctness:** 3
**Technical Novelty And Significance:** 2
**Empirical Novelty And Significance:** 2
**Recommendation:** 3

**Clarity, Quality, Novelty And Reproducibility:**

The paper is relatively clear and well-structured. Most, if not all, the hyper-parameters are reported in the appendix and it is easy to follow.



**Strength And Weaknesses:**

I find the concept of considering morphology and environment optimisation as MDP problems a very very interesting concept. I really like it and find this very interesting.
This is for me the main strength of the paper.

Unfortunately, the propsoed experiments are not enough to gauge the benefits of the proposed method. All the consdered baselines use fixed environment (as depicted in Fig1) without any possible curriculum learning. The introduction of the paper stresses clearly that co-evolving the morphology and the environment is essential. Yet, none of the considered baselines does this, while the paper cites several papers that co-evolve morphology and environment to create such curriculum learning. Typically, the proposed methods should be compared to POET[1] and Enhanced POET[2], which are both designed with the same objective and motivation than the proposed algorithms. Only with such comparison, we will be able to know if training policies to modify environments and morphology is a promising alternative.

[1]Wang, R., Lehman, J., Clune, J., & Stanley, K. O. (2019, July). Poet: open-ended coevolution of environments and their optimized solutions. In Proceedings of the Genetic and Evolutionary Computation Conference (pp. 142-151).
[2]Wang, R., Lehman, J., Rawal, A., Zhi, J., Li, Y., Clune, J., & Stanley, K. (2020, November). Enhanced POET: Open-ended reinforcement learning through unbounded invention of learning challenges and their solutions. In International Conference on Machine Learning (pp. 9940-9951). PMLR.

**Summary Of The Paper:**

In this paper, the authors propose to use three different policies to modify simultaneously the actions, morphology, and environment in a Reinforcement Learning problem.
While the control policy is very standard in RL applications, this paper also considers morphology and environment modifications as Markov Decision Processes. The state of the morphology considers some aspects of the robot's skeleton, while the state of the environment is some aspects such as roughness or dip length. The action spaces correspond to modifications of the state (of either the morphology or the environment). Finally, the reward functions are designed to encourage the agent to improve its learning progress and adaptability to varying environments.
The proposed algorithm is evaluated in 3 different tasks (2d-locomotion, 3d-locomotion, and gap-crosser). It is compared to two baselines (T2A and NGE. Finally, a series of ablation studies are presented to explain the inner workings of the proposed algorithms. In all the presented results, the algorithm outperforms the baselines.

**Summary Of The Review:**

Overall, the paper presents an interesting concept. Unfortunately, the selected baselines do not provide a fair comparison to evaluate the benefits of using MDP and policies for morphology and environment co-evolution.

---

> ### Author Response · Authors · 2022-11-18
> **Response to reviewer ugJV**
>
> Thank you for your suggestions and comments!
>
> ### Q1: All the considered baselines use fixed environment (as depicted in Fig1) without any possible curriculum learning
>
> - We use baselines that lack an environment-specific curriculum because our paper is the first work (to the best of our knowledge) studying to train an generalizable RL agent following a curriculum of varying morphology and environments.
> For a better and fair comparison, we modified the original baselines and train their models with periodically changing environments (Fig. 2). Moreover, we have  conducted a thorough ablation studies to demonstrate the effectiveness of our curriculum (Fig. 3(a),(b)).
>
> ### Q2: Comparison with POET and enhanced POET.
>
> - POET does NOT share the same goals and motivations as MECE, though they both developed certain curriculum. POET's motivations and objectives are **completely different** because:
>   - The main goal of POET is to train **an agent with a fixed morphology** to overcome the environmental challenges. In MECE, we aim to **evolve and optimize the agent's morphology** in order to make it **more adaptable to various environments**.
>   - In MECE, the agent’s morphology and the training environments are both evolving, while POET only changes the environment.
> Since the morphology is fixed in POET and not optimized for different environments, its learned RL policy is sub-optimal in terms of generalization to different environments when compared with the RL policy trained by MECE. For example, the length of the body or the number of the limbs may place a hard constraint that prevents the agent from crossing some obstacles or gaps, no matter what the control policy is.

---

> ### Author Response · Authors · 2022-11-28
> **Thank you for your review!**
>
> We sincerely appreciate your time and effort in reviewing our paper! We have made every effort to address your concerns, espically the main difference of the proposed method and other works.
>
> Hence, it would be highly appreciated if you could provide feedback to our responses or confirm whether there is no remaining concern. If you have any further concerns, questions, or suggestions, we are willing to discuss and reflect on them in the next revision.
> Thank you!

---

### Official Review · Reviewer_JRxx · 2022-10-29

**Confidence:** 4
**Correctness:** 2
**Technical Novelty And Significance:** 3
**Empirical Novelty And Significance:** 3
**Recommendation:** 5

**Clarity, Quality, Novelty And Reproducibility:**

Clarity needs to be improved by providing more formal math definitions.

Quality is good.

Novelty is also good since this paper deals with a new problem.

Reproducibility is unclear since it's not possible to re-produce the results for now because of no release of code.

**Strength And Weaknesses:**

Strength:

- The paper investigates a new problem of agent-environment co-evolution, which is interesting and valuable to the community.
- The method of using three separate policies for control, agent evolution and environment evolution is interesting and can be viewed as an extension of Transform2Act.
- The reward design for agent evolution policy and environment evolution policy is interesting, but it would be better to provide more explanation on these two.
- The experiment results are strong compared to previous methods.

Weakness:

- The biggest weakness is the problem definition itself. Though the problem formulation is interesting, I'm not sure if it is the right way to deal with the problem of robot agent learning. In practice, we would like to fix the distribution of environments but change the agent only:
  - Conceptually, "environment" represents the task itself and is the goal of agent design and learning.The method proposed in this paper essentially changes the task and makes the learning objective meaningless. Note that it is OK to evolve the agent because having better agents to work in the same task/environment is still meaningful. But it is not OK to evolve the environment because evolving/changing environment to make the task easier is cheating.
  - The authors may argue that the environment distribution with environment evolution policy is more diverse therefore better, as they argued in the first paragraph of Section 5.1. However, the final distribution of the environment has become uncontrollable, unless the author can show that the final distribution of the environments after environment evolution using policy is guaranteed to be exactly the wanted distribution. However, I cannot see that from the paper.
- Following up on the previous point, the algorithm may also have a problem. What replay buffer $\mathcal{D}$ contains is essentially an evolution trajectory of morphology $m_t$ and environments $\theta_t^E$ as suggested by Algorithm 1. This means the trajectory of morphology and environments in $\mathcal{D}$ cannot possibly have certain desired distribution, e.g. uniform distribution over a range. Moreover, the trajectory of morphology and environments is uncontrollable.
- The formal math notation definition of several variables are missing and makes the paper confusing. For example, what is the difference between $\mathcal{R}$ in Eq (1) and $\mathcal{R}^H_{\theta_{\beta t}^E}$ in Eq (2)?


**Summary Of The Paper:**

This paper proposes a new framework named MECE for the co-evolution of both agent morphology and environment. The core ingredient of the framework is three separate policies: one for controlling the agent, one for changing the agent morphology and one for changing the environment. The three policies are trained by exploration in the simulation environment. The experiments are done on three types of environments in simulation and the proposed method shows gain in terms of learning efficiency in terms of rewards compared to baseline methods.

**Summary Of The Review:**

Though the newly introduced problem is interesting, it is questionable whether the formulation is meaningful given the fact that the method essentially changes the task. Also, it is questionable whether the resulting task distribution can be controlled.

---

> ### Author Response · Authors · 2022-11-18
> **Response to reviewer JRxx**
>
> Thank you for your suggestions and comments! We respectfully disagree with you on "it is questionable ...essentially changes the task...questionable...resulting task distribution can be controlled". **The very aim of changing the training task is to build a curriculum that can train a morphology and a control policy generalizable to UNSEEN tasks and environments**, as we did in the evaluation phase. The experiments on UNSEEN environments show that our curriculum achieves much better generalization performance than previous works. We carefully address your major concerns as below:
>
> ### Q1:About the problem definition itself.
>
> - Making environments (tasks) easier is not a good curriculum for training a generalizable RL agent’s control policy and its morphology. MECE is not trained to reduce the difficulty of the environments. Instead, it aims to accelerate the evolution of the morphology when adapted to different environments:
>   - All our evaluation in this paper is conducted in environments randomly sampled from a uniform distribution that includes both easy and hard environments but MECE started from a highly easy (if not the easiest) environment. According to MECE’s outstanding performance in all evaluations, it is almost impossible that MECE keeps making the environment even easier than the initialized one.
>   - The design of rewards in our paper also prevents $\pi_e$ from making the environments easier because $\pi_e$ is trained to improve the progress of the morphology policy, i.e., to shorten the adaptation of the agent to a new environment. If MECE makes the environment easy, we will get zero progress on the RL agent and thus zero progress on the morphology policy. This is opposite to maximizing the accumulated progress.
>
> ### Q2: the distribution of the training environments
> - The goal of curriculum learning in this paper is to automatically generate a sequence of training environments in order to optimize the performance on possible test/unseen environments drawn from certain distribution. We design a reward $r_e$ for $\pi_e$, so it is optimized towards achieving greater progress on morphology optimization and control policy learning for any environment drawn from the distribution.
> WIthout a curriculum, it is not efficient to draw random environments or fix a single environment for training. In order to demonstrate the importance of our curriculum, we also carried out an ablation study (see Fig. 3 (a)). The findings demonstrate that generalization to various environments cannot be improved by training in randomly varying or a fixed environment (at least one type of environment).
>
> ### Q3: math notation definition of several variables are missing
> - Thanks for your advice! We have updated them in the new version.
>
> ### Q4: Reproducibility is unclear
> - We will release the source code when published.

---

> > ### Comment · Reviewer_JRxx · 2022-11-28
> > **Reply**
> >
> > I thank the authors for their response.
> >
> > I think the authors still did not understand the real problem of their algorithms. The real problem is their MECE and CO-EVO algorithms cannot guarantee the agent to be **trained** in morphologies and environments drawn from specific distributions (e.g. uniform distribution in the simplest case). The above reply from the authors still did not answer this important question.
> >
> > Let me make my point clearer here:
> >
> > According to algorithms proposed by the authors, the agent morphology and environment parameters $m_t$ and $\theta_t^E$ are updated as follows:
> > $$
> > \begin{split}
> > & m_{t+1} \leftarrow m_t, \theta_{t+1}^E \leftarrow \theta_t^E  \text{ (line 7 of Algorithm 1)}  \\\\
> > & (m_{t+1}, \theta_{t+1}^E) = \text{CO-EVO} (m_t, \pi, \theta_t^E) \text{ (line 11 of Algorithm 1)}
> > \end{split}
> > $$
> > and in the $\text{CO-EVO}$ algorithm, $m_{t+1}$ and $\theta_{t+1}^E$ are updated by
> > $$
> > \begin{split}
> > & \text{Apply } \pi_m \text{ to modify } m_{\alpha t} \text{ to } m_{\alpha t + 1} \text{ (line 6 of Algorithm 2)}  \\\\
> > & \text{Apply } \pi_e \text{ to modify } \theta_{\beta t}^E \text{ to } \theta_{\beta t+1}^E \text{ (line 10 of Algorithm 2)}  \\\\
> > \end{split}
> > $$
> >
> > So now it should be clear that two sequences $\\{m_t\\}_t$ and $\\{\theta_t^E\\}_t$ are not drawn from desired distributions but are essentially produced by applying $\pi_m$ and $\pi_e$ on $m_t$ and $\theta_t^E$ on the fly, which is not controllable at all.
> >
> > This is a very serious problem, because it is very likely that agents miss important samples from desired distributions during training. The trained policies will become highly unstable and random. Though the authors may luckily get their policies work on the evaluated environments and tasks shown in the paper, this issue still exists and will cause problems in some other scenarios.
> >
> > The above problem is fundamental and cannot be addressed by more explanations or more experiments. Therefore, I still vote for **rejection** of the paper. I hope the authors could address this problem in the next version of the paper by coming up with an improved version of the algorithm.

---

> > > ### Author Response · Authors · 2022-12-01
> > > **Thanks for the response but it is a misunderstanding of curriculum learning!**
> > >
> > >
> > > We appreciate your further clarification of the concerns in more detail! However, **it is a serious misunderstanding of the fundamental motivations of curriculum learning** [1,2]. The goal of curriculum learning is to select the most informative training examples or tasks for each training stage in order to improve generalization performance on unseen tasks (e.g., those drawn from uniform distribution). We respectfully disagree with several points you made in your latest comments, in particular:
> > >
> > > > “The real problem is their MECE and CO-EVO algorithms cannot guarantee the agent be trained in morphologies and environments drawn from specific distributions (e.g. uniform distribution in the simplest case).”
> > >
> > > - **There exist many well-known machine learning paradigms whose training distribution differs from the targeted one**, e.g., curriculum learning, transfer learning, off-policy RL, etc. Curriculum learning intends to avoid training on the uniform distribution because that is inefficient and non-adaptive to different learning stages.
> > >
> > > > “The two sequences … produced by applying $\pi_m$ and $\pi_e$ on the fly, which is not controllable at all”
> > >
> > > - **This is wrong! Both $\pi_m$ and $\pi_e$ are fully controllable and optimized for generalization to unseen environments** (e.g., ones drawn from uniform distribution). Their training is the main focus of our method and they are optimized to produce a better morphology that can be quickly adapted to new environments (by maximizing the agent’s progress), including those drawn from a uniform distribution.
> > >
> > > > “This is a very serious problem, because it is very likely that agents miss important samples from desired distributions during training.”
> > >
> > > -  **This is wrong because the very goal of our curriculum is to select the important samples bringing the greatest progress.** It is essential to understand that the **rewards to train $\pi_m$ and $\pi_e$ are learning progress-based and enable a more efficient exploration than uniform sampling**. Because progress on a fixed morphology and environment eventually approaches zero, $\pi_m$ and $\pi_e$ are optimized to switch to a new morphology and environment where the agent can make more progress. This leads to a much more efficient exploration of morphology and environment spaces than uniform sampling, which may select those with little progress (either too difficult or too easy).
> > >
> > > > “The trained policies will become highly unstable and random. Though the authors may luckily get their policies work on the evaluated environments and tasks”
> > >
> > > - **This is a groundless guess.** On the opposite, in experiments, we consistently observe a significant improvement over previous methods on different types of evaluation tasks with random, unseen environments. This is a much more challenging evaluation setting and less predictable than what is commonly used in RL (which uses the same task/environment as training). As a result, this is not by chance, but rather a solid improvement in generalization performance.
> > >
> > >
> > > 1. Remy Portelas, Cedric Colas, Lilian Weng, Katja Hofmann, and Pierre-Yves Oudeyer, Automatic Curriculum Learning For Deep RL: A Short Survey, IJCAI2020.
> > > 2. Sanmit Narvekar and Bei Peng and Matteo Leonetti and Jivko Sinapov and Matthew E. Taylor and Peter Stone, Curriculum Learning for Reinforcement Learning Domains: A Framework and Survey, J. Mach. Learn. Res., 2020[21], 181:1-181:50.

---

> > > > ### Comment · Reviewer_JRxx · 2022-12-01
> > > > **Reply**
> > > >
> > > > It is true that the core of curriculum learning is to select the most informative training examples or tasks for each training stage so that each training stage is much easier. However, no matter how the curricula are designed, the distribution of training examples or tasks of **the final stage** should match a desired distribution. I can't see that from the algorithm proposed by the author.
> > > >
> > > > > This is wrong! Both $\pi_m$ and $\pi_e$ are fully controllable and optimized for generalization to unseen environments
> > > >
> > > > This is a very interesting argument. The two policies $\pi_m$ and $\pi_e$ are the training objectives of the proposed algorithms, right? If they are fully controllable, why not just directly control them? Why still train them?
> > > >
> > > > As for "optimized for generalization to unseen environments", I can't see this from the algorithm either. Can the authors point me to which line(s) of their algorithms/equations are showing this? What I saw is that $\pi_m$ is used to update $m_t$, $\pi_e$ is used to update $\theta_t^E$, and that $\pi_m$ and $\pi_e$ are optimized for better training improvements. Why are $\pi_m$ and $\pi_e$ optimized for generalization to **unseen** environments?
> > > >
> > > > > This is wrong because the very goal of our curriculum is to select the important samples bringing the greatest progress. It is essential to understand that the rewards to train and are learning progress-based and enable a more efficient exploration than uniform sampling.
> > > >
> > > > **This is exactly where the problem is**. Please note that a curriculum that selects the important samples to bring the ***greatest
> > > > reward improvement*** does not guarantee that the samples can ***cover the enough space of the distribution***. These two are very different.
> > > >
> > > > In fact, selecting samples to get greatest reward improvement actually **hurts** sample coverage. Imagine those task samples that are difficult and low in reward. The $\pi_m$ and $\pi_e$ would be trained to avoid these samples because they bring bad reward improvements. However, these difficult task samples should actually be sampled with higher frequency than other easier tasks because more training is needed for the difficult tasks.
> > > >
> > > > Put it another way, selecting samples to get greatest reward improvement encourages $\pi_m$ and $\pi_e$ to stay in a "comfort zone" of easy tasks and hurts the training of control policy $\pi$ on hard tasks.
> > > >
> > > > > This is a groundless guess. On the opposite, in experiments, we consistently observe a significant improvement over previous methods on different types of evaluation tasks with random, unseen environments.
> > > >
> > > > Of course, as a reviewer, I don't have the code for this paper in my hand, so I cannot run the experiments myself and use the experiment results to support my comments. Therefore my comments will always be "groundless", so are other reviewers' comments.
> > > >
> > > > I hate to ask the authors to provide additional experiments. I think it would be beneficial for the authors to conceptually answer the following question:
> > > >
> > > > *If the task distribution includes both hard tasks (tasks that are likely yields low reward) and easy tasks (tasks that are likely to yield high reward), and $\pi_m$ and $\pi_e$ are trained for better reward improvements, how are $\pi_m$ and $\pi_e$ able to sample enough hard tasks?*

---

> > > > > ### Author Response · Authors · 2022-12-01
> > > > > **Can you please understand what is the "progress" we used for the rewards?**
> > > > >
> > > > > **You keep saying that $\pi_m$ and $\pi_e$ will stay with easy tasks. This is entirely WRONG!** As we stated many times, **we use progress as the reward** to train $\pi_m$ and $\pi_e$. **On easy tasks, there is no progress that can be made** because the control policy and morphology are already good at them! Moreover, an agent cannot keep making the greatest progress in a specific environment forever because there is a limit to the reward! So progress as a reward will prevent the policy from staying in its comfort zone, which is the opposite of your guess.
> > > > >
> > > > > If you are willing to read more details: to maximize the progress of the control policy, $\pi_m$ is enforced to generate a more adaptive and agile morphology on which the control policy's reward can be accumulated faster. Similarly, to maximize the progress of the morphology policy, $\pi_e$ is enforced to generate new environments in which the morphology evolution can make greater progress, i.e., modifying the morphology leads to faster adaptation of the control policy to the new environment. If the generated new environment is easy, the morphology and control policy already learned it and thus cannot make any more progress on it. To maximize progress, the environment won't stay easy and must change!
> > > > >
> > > > > **Selecting the hardest tasks can be problematic and terminate learning**: the agent can keep failing on those hard tasks and only receive zero rewards (no effective feedback), which is useless to the training. It is a BIG misunderstanding that the hardest tasks can provide the most information for policy learning.
> > > > >
> > > > > **Regarding optimizing $\pi_m$ and $\pi_e$ for generalization to unseen environments, this is also due to the progress-based rewards**: $\pi_e$ tends to generate an unseen environment because the progress that can be made in the trained environment is limited. $\pi_m$ is then optimized for faster adaptation to the unseen environment because its reward is the control policy's progress (the increase of the control policy's reward). You can find the precise definitions of progress-based rewards in Equ. (1)-(2) and their associated criteria are highlighted (in bold) in Introduction.

---

> > > > > > ### Comment · Reviewer_JRxx · 2022-12-01
> > > > > > **Reply**
> > > > > >
> > > > > > It is true that if $m_t$ and $\theta_t^E$ stay in the comfort zone of easy environments, the reward improvement or the so-called "progress" will eventually decrease to **zero**. I agree with that.
> > > > > >
> > > > > > The problem is, once $m_t$ and $\theta_t^E$ are in the comfort zone of easy environments and tasks, they will be refusing to go to harder environments and tasks, because when they do that, the reward improvement or the "progress" will be **negative** due to the drop of reward compared to easy tasks. But unfortunately, $\pi_m$ and $\pi_e$ are trained to maximize reward improvement or the "progress". So it is very unlikely that $m_t$ and $\theta_t^E$ can possibly jump out of the comfort zone of easy environments and tasks, due to the training objective of $\pi_m$ and $\pi_e$. This will hurt the sample coverage of the environments and tasks.
> > > > > >
> > > > > > > Selecting the hardest tasks can be problematic and terminate learning
> > > > > >
> > > > > > I never said that your algorithm need to sample the ***hardest*** tasks that return zero reward. As I mentioned before, one problem of your algorithm is that it is unlikely to sample enough hard but reasonable tasks once $m_t$ and $\theta_t^E$ enter a comfort zone of easier tasks.
> > > > > >
> > > > > > ------
> > > > > >
> > > > > > This is my last response to the authors' comments.
> > > > > >
> > > > > > In summary, I still believe that the algorithm proposed in this paper has serious problems that cannot be fixed by adding more explanations or more experiments. I'm not convinced by the comments and explanations from the authors either. Therefore, I still vote for rejection of this paper.

---

> > > > > > > ### Author Response · Authors · 2022-12-12
> > > > > > > **Staying in a comfort easy zone is opposite to our reward for $\pi_e$!**
> > > > > > >
> > > > > > > **Our curriculum would not stuck in a “comfort zone” because agents with different morphologies tend to make similar progress in such a comfort zone of easy environments**. In other words, in such a comfort zone, no matter how the morphology is modified, $\pi_m$ cannot not receive effective rewards or being improved — this is opposite to our training objective of $\pi_e$, which aims to maximize the accumulated progress of $\pi_m$. In contrast, in a “less comfortable zone” or a sufficiently challenging (but not too hard) environment, different morphologies tend to show different progress. As a result, using random exploration of the RL algorithm training $\pi_m$, our method can distinguish between good and bad morphology and thus achieve a significantly faster morphology evolution.
> > > > > > >
> > > > > > > At certain stages of our curriculum learning, there may exist hard environments with negative learning progress but this indicates that they are too challenging for the current stage and cannot improve the morphology or the agent at the moment. As Introduction pointed out, "a good training environment" should accelerate the evolution of morphology and maximize its progress, so negative learning progress indicates a bad training environment for morphology optimization.
> > > > > > >
> > > > > > > Though it has been stated numerous times, we would like to emphasize that **all of the above is primarily due to our novel design of the reward $r_e$ for training $\pi_e$**. This is the primary reason for our curriculum to consistently achieve much better generalization performance (no matter which random seed to use) than training with uniform random environments (which is a much poorer exploration of the huge environment space) and all other baselines, **not “luck”. Using learning progress as the reward would not make RL agents stuck in a comfort zone of easy environments. This has been demonstrated by a line of published works [1, 2].**
> > > > > > >
> > > > > > > [1] Rémy Portelas, Cédric Colas,Katja Hofmann and Pierre-Yves Oudeyer, Teacher algorithms for curriculum learning of Deep RL in continuously parameterized environments, CoRL, 2019.
> > > > > > >
> > > > > > > [2] David Held, Xinyang Geng, Carlos Florensa and P. Abbeel, Automatic Goal Generation for Reinforcement Learning Agents, ArXiv.abs/1705.06366, 2018.

---

> ### Author Response · Authors · 2022-11-28
> **Thank you for your review!**
>
> We sincerely appreciate your time and effort in reviewing our paper!
>
> We believe your constructive comments will further strengthen our paper, especially the math definition. In the response and revised paper, we have made every effort to address your concerns. Particularly,
> - A clearer explanation of the problem definition.
> - A more detailed analysis of the proposed method.
>
> Hence, it would be highly appreciated if you could provide feedback to our responses or confirm whether there is no remaining concern. If you have any further concerns, questions, or suggestions, we are willing to discuss and reflect on them in the next revision.
>
> Thank you!

---

### Official Review · Reviewer_9QtA · 2022-11-02

**Confidence:** 4
**Correctness:** 3
**Technical Novelty And Significance:** 3
**Empirical Novelty And Significance:** 3
**Recommendation:** 6

**Clarity, Quality, Novelty And Reproducibility:**

Quality:
The paper is good and clear.

Novelty:

The main idea is novel in this task.

Reproducibility:

The paper provides enough details to reproduce this paper.


**Strength And Weaknesses:**

Strength:

1. The proposed method is interesting and new, and may inspire more subsequent works in this area.

2. The presentation is clear and easy-to-follow.

3. The experimental results clearly demonstrate the supriority of the proposed method over baselines.

Weaknesses:

1. Although the proposed method only provides a quantitative evaluation of the learned morphology, a qualitative evaluation is also required for this task. Can the authors include some videos of how the agents perform in various environments?

2. The paper only demonstrates the performance of the proposed method when trained in a changing environment. Can the authors conduct some experiments on baselines trained in varying environments (even if they are random)? It is, in my opinion, compelling evidence for the efficacy of the proposed co-evolution method.

**Summary Of The Paper:**

This paper porposes an interesting method to learn the agent's morphology that are robust to the change of environments. To achive this, this paper proposes a co-evolution method to learn  two policies that automatically change the morphology and the environment, respectively. The experimental results demonstrate the supriority of the proposed method over baselines.

**Summary Of The Review:**

Overall, the proposed method is interesting and novel in the morphology design area. The experimental results also show the effectiveness of the proposed method.  The authors can provide more evidence in resolving my concerns provided in the "Weaknesses" section.

---

> ### Author Response · Authors · 2022-11-18
> **Response to reviewer 9QtA**
>
> Thank you for your suggestions and comments! We carefully address your major concerns as below:
>
> ### Q1: “videos of how the agents perform in various environments?”
> - Thanks for the great suggestion! We are going to generate the videos of the agents performing in various environments.
>
> ### Q2: conduct some experiments on baselines trained in varying environments
> - The original draft already includes the experiments you suggested, i.e., baselines trained in varying environments. The results are reported in Fig. 2, where  "Transform2Act" and "NGE" are trained in environments varying periodically and randomly sampled from a fixed distribution. They perform better than their original version trained in a fixed environment but still worse than our method.

---

> ### Author Response · Authors · 2022-11-28
> **Thank you for your review!**
>
> We sincerely appreciate your time and effort in reviewing our paper!
>
> We believe your constructive comments will further strengthen our paper. Hence, it would be highly appreciated if you could provide feedback to our responses or confirm whether there is no remained concern.
>
> If you have any further concerns, questions, or suggestions, we are willing to discuss and reflect on them in the next revision.
>
> Thank you!

---

### Decision · Program_Chairs · 2023-01-20

**Decision:**

Reject

**Justification For Why Not Higher Score:**

All reviewers found this to be a very interesting paper, but expressed a variety of concerns about the method and experiments. JRxx engaged in a long discussion on the soundness of the method, which the authors rebutted. The AC sides with the authors here and agrees that the general idea, in principle, could provide an effective teaching curriculum. However, this discussion does raise the issue of clarity: in the paper itself, it's hard to follow exactly how the rewards work and why they lead to a curriculum; in particular, Eqn. 2 appears to reward steps toward ever "easier" environments (perhaps there is a typo?) and does not clearly relate to morphology progress.

Another concern was with the baselines: ugJV suggested comparing to methods, such as POET, that evolve a curriculum of environements. The authors rebutted that prior such methods do not evolve morphologies. Nonetheless, the AC agrees that, while not entirely critical, it would improve the paper to at least attempt to compare to such methods. For example, POET could be adapted to also update the morphology of agents (e.g., using the morphology update algorithm from the current paper).

Finally there were lingering concerns that the paper is not as thorough as it could be, in terms of qualitative results and analysis of the method. Taking all these issues in sum, the AC feels that the paper is very promising but could use another pass on clarity, comparisons, and analysis before it meets the ICLR bar.

**Justification For Why Not Lower Score:**

N/A

**Metareview: Summary, Strengths And Weaknesses:**

Summary:
This paper presents an algorithm for co-evolving the control, morphology, and environment of an agent. The rewards are set up so that the environment provides a curriculum that trains the agent's morphology and control. The result is agents that behave robustly in held-out test environments.

Strengths:
* Co-evolving control, morphology, and environment is an interesting and novel idea
* Method convincingly outperforms baselines that do not co-evolve the environment

Weaknesses:
* No comparisons to baselines that do co-evolve the environment (like POET)
* The method is hard to follow and there was some disagreement about whether or not it is sound
* Hyperparameters may need to be tuned per environment
* More qualitative results and analysis of the learned behaviors, environments, and morphologies would be appreciated